# Research Progress in Energy Based on Polyphosphazene Materials in the Past Ten Years

**DOI:** 10.3390/polym15010015

**Published:** 2022-12-20

**Authors:** Zeping Zhou, Zhen Jiang, Feng Chen, Tairong Kuang, Dapeng Zhou, Fuliang Meng

**Affiliations:** 1College of Material Science and Engineering, Zhejiang University of Technology, Hangzhou 310014, China; 2Hangmo New Materials Group Co., Ltd., Huzhou 313000, China

**Keywords:** polyphosphazene, energy storage, supercapacitors, fuel cells, solar cells, lithium batteries

## Abstract

With the rapid development of electronic devices, the corresponding energy storage equipment has also been continuously developed. As important components, including electrodes and diaphragms, in energy storage device and energy storage and conversion devices, they all face huge challenges. Polyphosphazene polymers are widely used in various fields, such as biomedicine, energy storage, etc., due to their unique properties. Due to its unique design variability, adjustable characteristics and high chemical stability, they can solve many related problems of energy storage equipment. They are expected to become a new generation of energy materials. This article briefly introduces the research progress in energy based on polyphosphazene materials in the past ten years, on topics such as fuel cells, solar cells, lithium batteries and supercapacitors, etc. The main focus of this work is on the defects of different types of batteries. Scholars have introduced different functional group modification that solves the corresponding problem, thus increasing the battery performance.

## 1. Introduction

With recent rapid development of society and economy, the energy industry is developing rapidly. Advanced commercial electronic equipment such as new energy electric vehicles [1,2,3,4], mobile phones and other portable products, big data network systems, etc. [5,6,7], have an increasing demand for safe batteries with high energy density [8]. The scope of application also has higher requirements, such as temperature, acidity and alkalinity. Traditional fuel cells and lithium batteries can no longer meet such requirements [9,10]. This phenomenon greatly limits the development of large-scale energy storage equipment. Therefore, people have modified the electrodes, diaphragms and electrolytes of different types of energy storage devices and energy storage and conversion devices to improve their performance [11,12,13,14].

Polyphosphazenes are inorganic-organic polymers with a long linear backbone which comprise alternating nitrogen and phosphorus atoms [15,16]. Compared with other hydrocarbon long-chain polymers, it has many advantages. First, the stable dΠ-pΠ conjugation formed by the nitrogen and phosphorus structures of the main chain contributes to the chemical stability of the polyphosphazenes, while the softer chain structure contributes to their thermal stability and excellent film-forming properties [17,18]. Secondly, the side group chlorine atom beside the main chain is highly reactive, and its side chain group can be easily changed by physical and chemical modification to improve various properties [19]. Due to its unusual structure and properties, polyphosphazene has become a research hotspot in various fields, including slow drug release, bone tissue regeneration, solid electrolytes, supercapacitors, etc. Quite good results have been achieved in these fields [14,20,21]. This article focuses on the modified application of polyphosphazene materials in various energy storage devices and energy storage and conversion device in the past ten years.

## 2. The Synthesis of Polyphosphazenes

The first linear polyphosphazenes were synthesized through thermal ring-opening polymerization of hexachlorocyclotriphosphazene (HCCP) at 250 °C under a vacuum environment by Allcock et al. in 1965. However, the polymerization was scarcely controlled and the obtained product, polydichlorophosphazene (NPCl_2_)_n_, exhibited wide molecular weight with n ranging from 3 to 15,000 [22,23]. To obtain polymers with narrow molecular weight distributions, Allcock et al. [24,25] introduced PCl_5_ to serve as an initiator in the living cationic polymerization of phosphoranimines accomplished at ambient temperatures (ca. 25–35 °C), offering the possibility to obtain di-or tri-block organic macromolecules [23] with various structures such as star, comb and dendritic structures [23,24,25,26,27]. Later, Potts et al. [28] discovered that polymerization could occur in a solution with a high boiling point using BCl_3_ catalyst, which has many advantages such as lower temperature, shorter reaction times, and controllable reaction rates. However, this method is not commonly used because it requires precise temperature control, and excessive BCl_3_ may cause the molecular weight to decrease [28,29]. At the same time, excluding the most basic ring-opening polymerization of HCCP, there are also four main methods: catalytic solution polymerization [30,31], controllable room temperature active polymerization [32,33,34], condensation of n-dichlorophosphonyl-p-trichloromonophosphonitrilene (C1_3_P = NP(O)C1_2_) and one-step method by using NH_4_Cl and PCl_5_ as monomers (${\mathrm{PCl}}_{5}{+\mathrm{NH}}_{4}\mathrm{Cl}\;\to {\left(\mathrm{NPCl}\right)}_{n}+4\mathrm{HCl}\;(n=3–9)$) [35,36,37,38]. After multiple recrystallization and collection of trimer, linear polydichlorophosphazene was synthesized by ring opening thermal polymerization at 250 °C in a vacuum environment [35].

In addition, derivative synthesis methods and direct synthesis methods are also used for the preparation of linear polyphosphonitrile. The former is the preparation of new linear polyphosphonitrile by nitrification, diazotization and other polyphosphonitrile derivative reactions, while the latter is a polyphosphonitrile prepared by polymerization of small molecular monomers under certain conditions [39]. Presently, the most mature and commonly used method to prepare linear polyphosphonitriles is polymerization, followed by replacement.

Although linear polyphosphazene shows potential application prospects in many fields due to its molecular diversity and has many advantages that cannot be replaced by organic polymers compared with organic silicon, the biggest problem of this organic-inorganic hybrid polymer is that it is difficult to prepare high molecular weight polyphosphazene with low yield, which limits its practical application. Compared with polydichlorophosphazene, the preparation of hexachlorocyclotriphosphazene, a small molecule phosphazene compound, is much easier [19].

## 3. The Application of Polyphosphazenes in Electrochemistry

Due to the large number of functional groups that can replace active chlorine atoms on polydichlorphosphonic nitrile, the linear type shows a strong molecular design ability. Figure 1 shows various hydrolytically stable linear polyphosphazenes obtained by replacing polydichlorophosphazenes with different types of nucleophiles. The variety of side groups of linear polyphosphazene endows it with rich physical and chemical properties. Such variety in the physical and chemical properties gives them important application value in many application fields. In the past decades, a large amount of literature have reported the application of linear polyphosphazene in special rubber and elastic materials [38,39,40,41], fire retardant materials [42,43,44], biomedical materials [45,46], nonlinear optical materials [47,48], separation membrane materials [49,50], doped materials [51,52,53,54], solid polymer electrolytes [55,56,57,58,59,60,61], proton conducting polymer materials [62,63,64], catalyst carriers [65,66], polymer dyes [67] and other fields.

At the same time, due to its good thermal stability and flame retardancy, and since the six chlorides connected to the phosphorus atom have similar chemical activity to the linear polyphosphazene, the chemical research regarding phosphazene based on hexachlorocyclotriphosphazene has gradually become known and attracted great attention from researchers [68,69]. Figure 2 shows the synthetic schemes of the phosphazene derivatives. Hexachlorocyclotriphosphazene has six active sites, so it is often used as a central core to design and synthesize star shaped polyphosphazenes or dendritic polyphosphazenes. In addition, other multifunctional molecules can be selected as the central core, and linear polyphosphazene can be grafted as the branch chain to prepare star shaped or dendritic polymers [70].

The synthesis of cyclophosphnitrile does not need to be controlled by asymmetric substitution. Instead, the multi-functional cyclophosphnitrile reacts with monomers with bifunctional or multi-functional groups, resulting in a highly cross-linked reticular polymer structure. The main types of synthesis reactions are addition, condensation, metal complexation and Friedel-Crafts reactions. However, due to the highly cross-linked chemical structure of the cyclocrosslinked polyphosphonitrile, it is usually an insoluble resin, which is difficult to process. Therefore, at present, the application of cyclocrosslinked polyphosphonitrile is mostly concentrated in the fields of thermosetting resin, adhesives and polymer substrate filling material, which use the high content of phosphorus and nitrogen elements to improve the thermal stability and flame-retardant properties of the polymer matrix [42,43,44,45].

Polyphosphonitrile materials usually have special properties of template-induced self-assembly, which gives them a relatively excellent ability to cover different materials. By adding an external template during the polymerization process or using the TEA·HCl template formed by itself, polyphosphonitrile can form a coating with a highly cross-linked structure outside the template [71,72]. Due to the existence of lone pair electrons in the N and P elements in the polymer chain, it is easy to carry out complex coordination with the transition metal ions, and then insert a layer of transition metal nanoparticles on the surface of the polymer material. At the same time, some groups rich in the polymer surface have the potential to react with other functional groups, which can further modify the polymer surface. The molecular structure of polyphosphonitrile material contains a large number of heteroatoms such as O, N, P and S [73,74,75]. As a new precursor material, porous carbon materials with heteroatomic doping structure and high specific surface area can be prepared by carbonizing it at high temperature [76,77,78]. These unique properties are different from common carbon precursors such as organic small molecules, linear organic polymers and polysaccharides, which can be used as electrode materials and substrate materials of various catalysts [79,80,81].

Therefore, combined with preliminary research results, this article mainly introduces the application of polyphosphazene application in fuel cells, lithium batteries, solar cells and electrocatalysis.

### 3.1. Fuel Cells

Fuel cells operate on the same principle as conventional batteries, in which they transform chemical energy to electrical energy via an electrochemical redox reaction between the fuel and oxidant [82]. They can be divided into five types according to the different electrolytic components, namely, SOFC (solid oxide fuel cell), AFC (alkaline fuel cell), MCFC (molten carbonate fuel cell), PEMFC (proton exchange membrane fuel cell) and PAFC (phosphoric acid fuel cell) [83]. AEMFCs (anion exchange membrane fuel cells) and PEMFCs have received extensive research [84,85]. However, practical applications of PEMFCs are determined by many factors, including proton exchange membrane (PEM), which is considered the most critical factor affecting PEMFC performance [86]. Nafion membranes, one kind of PEM, are widely used due to their good proton conductivity, stable electrochemical performance and high mechanical strength throughout prolonged working times [87]. However, some questions are still a concern for its widespread use in commercial applications, including its high manufacturing costs, high fuel permeability, low working efficiency in the humidity and high temperature (above 80 °C) environment [88,89,90,91]. To promote extensive PEM application, numerous efforts have been made to reduce manufacturing costs and improve the conductivity and stability of the membrane. At present, research on PEM materials mainly focuses on polymer proton exchange membrane [92,93,94,95], ceramic proton exchange membrane [96,97,98] and organic-inorganic composite proton exchange membrane [99].

Polyphosphazenes are inorganic in nature and are superior to organic polymers. Firstly, they are flexible because of the –P=N– backbone, have no conjugated-based structure and no restriction upon free rotation and can exist in isomeric forms. Secondly, due to the unique properties and easy modification, using polyphosphazene polymers as PEMs can solve problems regarding membrane stability. Simultaneously, various side-chain substituent groups can be modified to achieve increased proton conductivity and improved efficiency. Herein, we introduce some applications of polyphosphazene polymers in PEMs in the past ten years.

Wang and coworkers mixed sulfonated phenoxy polyphosphazene (SPOP) and mPBI and the cross-linking agent polyfunctional triglycidyl isocyanurate (TGIC) [100]. The obtained mPBI-TGIC-SPOP composite membrane exhibits good mechanical potential and dimension stability along with oxidation resistance, contributing to a 3D network that is cross-linked. The research results demonstrate that conductivity of the studied mPBI-TGIC-SPOP composite membranes increased with increasing temperature, RH, as well as the amount of SPOP. Compared with other types of polybenzimidazole (PBI) membranes, mPBI-TGIC (5%)–SPOP (50%) membrane’s proton conductivity reached 0.143 S/cm (100% RH, 180 °C), which almost reached the highest value. In addition, the mechanical strength of membrane was 21.9 MPa, which could meet the PEM criterion. Moreover, mPBI-TGIC (5%)–SPOP (50%) composites had low permeability of methanol (7.95 × 10^−8^ cm^2^/s), but good selectivity of the membrane (6.08 × 10^5^ S/cm^3^) at 60 °C, positing a promising utilization in direct methanol fuel cells.

Santoshkumar and his coworkers introduced SPOP into SPEEK to prepare acid-based blend membranes [18]. The membrane had better mechanical performance and higher ionic conductivity compared with pristine SPEEK. Additionally, when cells were polarized, SPOP-SPEEK membrane attained a power density peak of 935 mW/cm^2^, consistent with that of Nafion—212 membrane.

Bozkurt and colleagues synthesized a series of triazole-containing polyphosphazene PEMs [101]. They first substituted 4–methylphenoxy with one chlorine atom on the side chain to obtain poly(4–methylphenoxy)–chlorophosphazene (PMPCP). The remaining chlorine atoms were then substituted with 3–amino–1H–1,2,4–triazole and 1H–1,2,4–triazole, yielding two triazole-containing polyphosphazene materials that were then reacted with trifluoromethanesulfonic acid to yield the corresponding triazole-containing polyphosphazene PEM (Figure 3).

According to experimental results, trifluoromethanesulfonic acid might enhance the membrane’s proton transferability, leading to proton hopping from N–H site to sulfate ions, thereby promoting proton conduction.

Zhu and his coworkers prepared an array of composite membrane CF_3_–PSx–PSBOSy–SCNT of copolymer poly[(4–trifluoromethylphenoxy) (4–methylphenoxy) phosphazene]–g–poly{(styrene)x–r–[4–(4–sulfobutyloxy)strene]y} (CF_3_–PSx–PSBOSy) doped with S–SWCNT (Figure 4) [102].

According to research results, the membrane had high proton conductivity as well as low methanol permeability, resulting from S–SWCNT. In addition, at 100 °C, in fully hydrated conditions, CF_3_–PS_11_–PSBOS_33_–SCNT along with CF_3_–PSBOS_45_–SCNT exhibited proton conductivities of 0.46 and 0.55 S/cm, respectively, which were 2.2–2.6 times that of Nafion 117. The comparative study table of the performance were shown in Table 1.

However, although PEMs have gained widespread attention because of their high-power density and prolonged use, they also have drawbacks such as high cost of precious metal catalysts and high fuel penetration, which limit their utility [106,107,108]. AEMFCs have recently developed rapidly and have higher electrode reaction activity than PEMFCs. In addition, anion exchange membrane (AEM) uses a non-noble metal catalyst and has lower fuel permeability [109,110]. As one of the essential components of AEMFCs, AEM is critical in the anion transport process. The ideal AEM should have high ionic conductivity and excellent alkali resistance, along with good mechanical characteristics and thermal stability [111,112]. However, compared to PEM, AEM exhibits lower ion conductivity and poor stability [113,114]. The researchers have addressed these two issues by identifying suitable cationic functional groups and developing a more robust polymer backbone.

Currently, the research on the main chain structure of basic AEMs focuses primarily on polysulfones [115,116,117], polyphenylene ethers [118,119,120], polybenzimidazole [121,122], polyetheretherketone [123,124] and polyphosphazenes [125]. These structures can improve their mechanical and electrical conductivity to a certain extent. Here we mainly introduce some research on AEM with a main chain structure of polyphosphazene.

Han et al. synthesized a series of tetraphenylphosphonium cations via nickel-catalyzed coupling reactions to prepare the corresponding anion exchange membranes [103]. This membrane minimizes electrostatic attraction by increasing electron density of the cation site. This is pivotal for enhancing OH-conductivity and alkaline stability of AEMs. As a result, the prepared PPMPPs exhibited excellent alkaline stability and satisfactory ionic conductivity in high pH environment under elevated temperatures.

Chen et al. introduced crown ethers in place of conventional quaternary ammonium salt groups into polyphosphazenes to prepare polyphosphazene AEMs with different crown ether contents (Figure 5) [50]. The experimental results demonstrate that, as the level of crown ether increased, the ionic conductivity of membrane would also increase. This was because the crown ether and sodium ions combined to form an efficient ion transport channel in the membrane, allowing hydroxide ions to freely move within it. Simultaneously, crown ether addition increased the water storage capacity of membrane, reducing OH- resistance to transport in the membrane, thus elevating the membrane ionic conductivity. When the crown ether content reached 60%, the ion conductivity of AEM was 78.6 mS/cm at 90 °C, and after soaking in 4 mol/L NaOH for ten days, the degradation rate of ion conductivity was 2.5%. After immersing in 2 mol/L NaOH for 1000 h, the degradation rate of ionic conductivity was 0.99%, indicating that crown ether addition fundamentally solved the alkali stability issue of membrane and could effectively improve AEM durability. Simultaneously, the membrane exhibited good thermal stability, low water absorption rate, low swelling rate and good mechanical properties, which met AFC requirements.

Zhu et al. [126] prepared a series of imidazole group polyphosphazene AEMs by mixing the prepared brominated poly[bis(4–methyl–phenol)phosphazene] with N-methylimidazole and LDH in DMF solution (Figure 6). The experimental results reveal that LDH addition could significantly improve the ion conductivity of AEM. The mixed membrane prepared under the current field had higher ion conductivity, which was higher than 18 mS/cm at 80 °C. However, without introducing LDH, the maximum ionic conductivity of membrane was only about 6 mS/cm.

In general, the researchers successfully exploited polyphosphazene’s superior skeleton and easily modifiable structural properties to provide fuel cell membranes with key properties such as good electrical conductivity, good mechanical properties and good fuel permeation resistance. This provides a good direction for the development of fuel cells. Apart from membrane applications, it is also applicable in catalysis.

Wu et al. used HCCP and 2-nitrobenzene-1,3,5-triol to prepare a novel kind of poly[cyclotriphosphazene-co-1,3,5-triol nitrobenzene] (PCTNB) microspheres [127], and was used to wrap carbon nanotubes to prepare PCTNB@CNTS composite materials through a series of high-temperature carbonization and potassium hydroxide activation steps to generate a highly active catalyst, which exhibited high catalytic activity, good methanol resistance and good stability (Figure 7). The reported degradation in the ORR polarization curves’ half-wave potential for PCTNB@CNTS was only 15 mV after 10,000 potential cycles, which was much smaller than commercial Pt/C catalyst (which was reduced by nearly 40 mV). Since the starting potential and half-wave potential were higher than that of other non-metallic electrocatalysts, it was expected to be an efficient, as well as durable ORR catalytic material in actual acid fuel cells.

To solve problems of poor durability and poor anti-CO poisoning ability of ethanol oxidation electrocatalysts in ethanol fuel cells, Yu et al. used SiO_2_ as a template to coat polyphosphazene (PCCP) on its surface through in-situ polymerization [128], followed by high-temperature calcination and then hydrofluoric acid attack to form a porous hollow material with a high specific surface area doped with N, P, S heteroatoms. The required catalyst was prepared after supporting Pd (Figure 8). The catalyst demonstrated high catalytic activity and cycle stability and could reach a high-quality peak current density of 1686 mA/mg, which was 2.8-fold that of Pd/C. Simultaneously, after 3600 s of durability test, its stability was also better than that of Pd/C catalyst. This was because the material possessed a high specific surface area, which facilitated electrolyte diffusion and charge transfer, and N, P, S heteroatoms would have a synergistic effect with Pd to improve electrochemical activity. It had a wide range of potential applications in ethanol fuel cells.

### 3.2. Li-Ion Batteries

As a new kind of precursor material, the molecular structure of polyphosphonitrile material contains a large number of heteroatoms, such as O, N, P, S. After carbonization at high temperature, porous carbon materials with heteroatom doped structure and high specific surface area can be prepared. These unique properties are different from common carbon precursors such as organic small molecules, linear organic polymers and polysaccharides.

Tao et al. prepared the N, P and S doped dual carbon-confined Fe_3_O_4_ nanospheres (Fe_3_O_4_@C@G) by the multi-heteroatom-doped dual carbon-confined strategy [73]. As expected, the resulting Fe_3_O_4_@C@G can serve as universal anode materials in lithium/sodium-ion batteries (LIBs/SIBs). Zhang et al. used a simple precipitation polymerization method to form polyphosphazene on Si particles to introduce a carbon layer to overcome the inherently poor conductivity and significant volume change [78]. With core-shell composite microspheres, Si anodes exhibit high cycle stability and magnification performance. In addition, the coulomb efficiency of 99% is still maintained after 100 cycles.

Zhou et al., through a simple hydrothermal method, obtained polyphosphazene with hierarchical MoS_2_ nanoflakes on the surface [129]. The obtained hollow carbon microsphere structure had not only high electrical conductivity, but also showed strong structural stability. These composites had a high reversible capacity (1421 mAh/g), excellent stable cycling performance and better rate capability.

Zhou et al. used a simple precipitation polymerization method to prepare a special structure used molybdenum disulfides as a core, polyphosphazene as a shell layer and 3D carbon network as a link, without any auxiliary template [51]. Electron chemical tests showed that the composite microspheres used a flexible conductive material to seal the active material, retain part of the pores to promote expansion and contraction. The composite microspheres could fully release the expansion stress of the active material and had stable electrical contact, thereby greatly improving the electrical properties of the material. At the same time, simple first-principles calculations showed that N, P, S doped modified carbon structures can improve the binding energy of materials and lithium ion/atoms and change the local charge density of materials to produce more active sites.

Zhang et al. produced the fabricated material of Si/C core shell through coating with polyphosphazene via a method of facile precipitation polymerization to address the intrinsic low electrical conductivity [130] as well as the large volume change. The results (Figure 9) showed that the amorphous carbon increased the interface reaction kinetics and electrode conductivity, leading the Si anode to show elevated cycling stability (740 mAh/g persisted after 100 cycles) along with rate capability (300 mAh/g at a rate of 2C).

F.F. Steward’s groups showed a new way to obtain the lightly packed cyclic phosphazenes as anode materials for Li-ion batteries [131]. With this method, we substituted HCCP with mono-functional materials. We then further chemically cross-linked it with other agents, generating a 3D network structure to provide dimensional stability and porosity for lithium ions diffusion. Given its intrinsic electronic insulation, the cyclic phosphazene material, in combination with 10 wt% graphite, produces 183 mAh/g capacity after 50 cycles.

GS Pappas et al. conducted a lithium-ion battery test on the PZSs-derived carbon nanospheres obtained after heat treatment at 850 °C [132]. Carbon spheres with high heteroatoms (N, P, S, O) doping, an average diameter of 500 nm, a high distinct surface area of 875 m^2^/g and a layered porosity were generated. The first discharge capacity exceeds 1126 mAh/g, which is irreversible. The capacity is approximately 440 mAh/g, exhibiting an active surface that triggers electrolyte decomposition along with formation of SEI on the anode. Due to the unique porous structure, doping of heteroatom and high surface area have synergistic effects. As an anode material, it can carry out long-term charge and discharge cycling of up to 1000 cycles at a 1C rate, showing remarkable stability and coulombic efficiency. The specific capacity can deliver about 130 mAh/g.

Zhou et al. prepared a core-shell structure material with cyclomatrix poly(organophosphazene) (POP) nanospheres serving as the primary material and bi-ZIF as the shell [133]. Among them, POP is a covalently cross-linked framework with high heteroatom content prepared by a one-pot method (Figure 10), and conditions can be further changed to obtain porous carbon spheres exhibiting well-preserved morphology along with heteroatom doped in situ. The presence of cobalt, as well as nitrogen, improves the carbon electrode surface area, as well as the wettability, enhancing the transport of protons and electrons. The formed core@shell carbon nanospheres have hierarchically meso-porous structures. A carbonized covalent inorganic-organic polyphosphazene framework harboring in-situ doped heteroatoms like N, P, S and O formed the origin of the carbon core. A bi-ZIF, containing up to 40% Zn, as well as Co elements, formed the origin of the shell structure. The surface area of BET of the carbon shell derived from bi-ZIF is 1348 m^2^/g, the Langmuir surface area is 1883 m^2^/g, and the total surface area of BET of the core@shell structure is 1025 m^2^/g. When used as a negative electrode material for lithium-ion batteries, the carbon structure core-shell has a charge capacity of 595 mAh/g and a discharge capacity of 546 mAh/g. After 140 cycles, the reversible charge/discharge capacity remains at 400 mAh/g, which is higher relative to the theoretical capacity of the graphite anode. Good cycle stability was obtained and 83% of the capacity was maintained in the C rate test.

A polyphosphazene/GN nanocomposite was readily fabricated via thermal polymerization of hexachlorocyclotriphosphazene with graphene oxide, and exhibits stable, as well as uniform nanostructure [134]. The addition of electronic-withdrawing poly-phosphazene nanoparticles on the graphene allowed the efficient creation of numerous active sites and avoidance of aromatic graphene layers stacking. This resulted in more domains for storage of lithium and accelerated transformability of charge. As a metal-free Li-ion battery anode, an excellent rechargeable capability (1002 mAh/g) along with rate performance (321 mAh/g at 5 A/g) could be achieved.

When discharging or charging existing inorganic lithium transition metal oxide cathodes at high current density, mass transfer limitations will elevate the temperature of the battery. This not only results in safety problems, but also reduces the energy density and cycle life of the battery. In addition, the inorganic lithium transition metal oxide cathode used in rechargeable batteries contains toxic heavy metals. Therefore, free radical polymers with good redox activity and stability have been proposed as alternative materials for inorganic metal oxide cathodes.

Serkan et al. developed a new synthetic strategy (Figure 11) for polyphosphazene harboring stable nitroxide radicals as a pendant group using sodium 4–bromophenoxide and polydichlorophosphazene to prepare polybis(4–bromophenoxy)phosphazene [135]. Then, n-BuLi is added along with 2–methyl–2–nitrosopropane dimer into a Schlenk tube to obtain polybis(4–(N–tert–butyl–N–oxylamino)phenoxy)phosphazene. This was generated via the silver oxide treatment of the correspondent arylhy–droxylamine polyphosphazene, Ag_2_O, serving as an oxidizing agent in benzene. Polybis(4–(N–tert–butyl–N–nitroxide) phenoxy) phosphazene was produced. An EPR (electron paramagnetic resonance) experiment was conducted to verify the generation of oxygen–based nitroxide unpaired electrons. The resultant material was explored as a cathode–active component for rechargeable lithium–ion batteries that perform at a 80 mAh/g capacity with a C/2 current density over 50 cycles.

Serkan et al. have fabricated a polyphosphazene containing stable nitroxide radicals with a theoretical capacity of 186 mAh/g [136]. A new polyphosphazene harboring pendant stable nitroxide aromatic radical groups bearing four electrons engagement per repetitive unit was fabricated. To accomplish this, they performed a sequence of macromolecular substitution reactions of poly (dichlorophosphazene) with 3,5–dibromophenol, 2–methyl–2–nitrosopropane along with lead oxide (Figure 12A). The cell exhibited a good rate of performance, harboring a discharge capacity of 100 mAh/g at a C/2 current density over 500 cycles (Figure 12B). The CV of PNPP demonstrates a well-defined oxidation and reduction couple (E_1/2_ = 3.62 V vs. Li/Li^+^) resulting from nitroxy radical oxoammonium cation generation during anodic scan and subsequent reverse formation in the cathodic scan. The peak separation of the redox wave is less than 70 mV at a relatively narrow 0.1 mV/s sweep rate. It could reflect the rapid transfer of electrons of the nitroxy radicals on polyphosphazene backbones.

As a lithium-ion conductor, electrolyte materials are an indispensable part of lithium-ion batteries. In order to solve battery safety issues, solid electrolytes are favored by more and more people and have become an ideal solution for the formation of novel lithium-ion batteries. Presently, the most researched electrolyte materials mainly include organic polymer solid and inorganic solid electrolytes [137]. Although the room temperature conductivity of the inorganic solid electrolyte can be an order of magnitude higher than that of the organic polymer-based solid electrolyte, the problem of the existing interface between the electrolyte and the electrode material severely limits its application. Polymers are considered to be a solution to the problem of electrolyte interface because of their special structures. However, because of the regular arrangement and crystallization of the polymer chain segments, the room temperature conductivity of polymer-based solid electrolytes can only reach 10^−6^–10^−8^/cm, which is far from meeting the requirements of commercial use. Therefore, it is a feasible solution to modify the polymer to reduce its crystallization behavior through molecular structure design.

In 1984, Allcock et al. modified linear polydichlorophosphazene (PDCP) and grafted diethylene glycol monomethyl ether to the side chain of PDCP through a nucleophilic substitution process to prepare poly (diethylene glycol monomethyl ether) phosphazene. Its glass transition temperature (Tg) is about 80 °C and the room temperature conductivity can reach a level of 10^−4^/cm. Since the discovery of poly[2-(2-methoxyethoxy)ethoxy–phosphazene] (MEEP), MEEP has not been studied and applied on a large scale. The reason is mainly because its lower glass transition temperature leads to poor mechanical performance, and viscous flow is prone to occur at high temperatures, leading to short-circuiting of the positive and negative electrodes. In order to solve the shortcomings of the poor mechanical properties of polyphosphazene-based solid electrolytes and make full use of its higher room temperature conductivity, researchers have devoted themselves to modifying its molecular chain structure or blending it with other materials and a variety of high-performance electrolytes.

Schmohl et.al using a mixed substitution of phosphorus consisting of 2-(2-methoxyethoxy)ethoxy side groups and anionic trifluoroborate groups to increase the low lithium ion conductivities of the conventional lithium salt containing MEEP [80].

Yu et al. reported a layered hybrid solid electrolyte fabricated through using a protective polymer electrolyte to coat a ceramic LATP electrolyte [81]. Polyphosphazene/PVDF-HFP/LiBOB, highly ionic conductive oligoethylene oxide functionalized polyphosphazene, MEEP and highly mechanically stable poly(vinylidene fluoride-co-hexafluoropropylene) (PVDF-HFP) were rationally chosen to fabricate SPE as a polymer matrix to ensure eligible electrochemical characteristics, as well as interfacial mechanical potential. Mobile lithium ions in the polymer layer are provided by LiBOB (Figure 13A). As shown in Figure 13B–F, fitted with a polymer and suitable ceramic constituents, the hybrid electrolyte exhibits favorable characteristics, including a flexible surface, high chemical stability, as well as high ionic conductivity and extensive electrochemical stability window (4.7 V). These properties will aid to realize its utilization in all-solid-state lithium batteries. The generated all-solid-state battery with a metallic lithium anode, along with high-voltage Li_3_V_2_(PO_4_)_3_/CNT cathode, exhibits high capacity as well as remarkable cycling performance and insignificant capacity loss over 500 cycles at 50 °C.

### 3.3. Other Energy Storage

Over the last decade, the above-mentioned polyphosphazenes have made significant research progress in lithium batteries and fuel cells, yielding promising results that can help to boost the shortcomings of corresponding batteries. Simultaneously, as the energy industry emerges, supercapacitors, solar cells, etc., are increasingly entering the public’s awareness [138,139,140,141,142,143]. Polyphosphazenes exhibit corresponding applications in these two areas due to their unique advantages.

As science and technology advance and people’s living standards improve, solar cells have become the most popular kind of green energy in the future. The perovskite solar cell, as a third-generation solar cell, has advanced rapidly in recent years. In 2009, Miyasaka et al. reported the first perovskite-type solar cell with a power conversion performance of approximately 3.8% [144]. However, in the following few years, by optimizing film growth, interface characteristics and absorbent composition, the power conversion rate increased to 23.3%. While polyphosphazene is a good material, Hailegnaw et al. [145] prepared a mixed-cation mixed-halide perovskite solar cell by introducing a phosphazene derivative between the electron-transporting layer and the back metal-contact as its buffer layer. They tested its optical performance and light stability. From the results, it had a higher power conversion efficiency of 17.3% and a 23.5 mA/cm^2^ short-circuit current density, whose fill factor was 72%. The PPz buffer layer enhanced photovoltaic activity and resulted in longer charge carrier lifetimes in the cells. When applied in organic solar cells as an electron-transporting interlayer, it improved electron extraction at the cathode, enabling the PPz interlayer to be extended to use different metals (aluminum, gold, copper, and silver) as the top contact in the generated PCSs.

Apart from solar cells, polyphosphazene has also been employed in energy storage. Polyphopronile-derived mesoporous carbon nanoparticles have a layered porous structure and highly doped heteroatoms (N, P, S, O), which allow them to induce additional Faraday redox reactions to improve the pseudocapitance properties of carbon-based electrodes. Microporous and mesoporous structures can facilitate the penetration and transport of electrolyte ions and favor electrochemical properties at high current densities. Chen et al. synthesized the hybrid hollow carbon precursor by a one-step rapid polycondensation reaction and encapsulated the supramolecular vesicles with polyphosphazene [82]. The high surface area and high heteroatom level of HCMS contribute to their excellent capacitive performance. The hollow carbon structure improves the utilization efficiency of surface area and realizes the short diffusion path of electrolyte ions. The specific capacitance in 6 M KOH electrolyte is 314.6 F/g when the current density is 0.2 A/g and 180.0 F/g when the current density is 30 A/g. The capacity retention rate after 2000 cycles is 98.2%.

Yang et al. reported the synthesis of a new cross-linked polyphosphazene through the reaction of 4,4′–diphenylmethane diisocyanate (MDI) and poly–bis(4–carboxyphenyloxy) phosphazene (PBCP) to study the relationship between the structure of carbon and the different weight ratios of PBCP/MDI [139]. Qiu et al. prepared a series of ultra-high surface area carbon materials with multi heteroatom doping by pyrolysis of poly (bisphenoxy) phosphazene (PBPP) through a simple chemical foaming strategy [141]. These materials have high specific capacitance in 6 M KOH because of the ultra-high specific surface area and wide hierarchical pore structure.

Wu and his coworkers prepared a novel carbon material with heteroatoms for use as supercapacitor electrodes through carbonization and activation of polyphosphazene precursor [138]. Their results show that the carbon material had a high specific surface area, especially when prepared at 800 °C, with a 1798 m^2^/g surface area with high porosity. Additionally, it demonstrated good cycling stability, retaining 89.5% capacitance over 10,000 cycles, which was attributed to the N, P, S heteroatoms, which improved the electronic conductivity of carbon materials, their wettability to electrolyte solution and their degree of graphitization. However, when activated at varying temperatures, a range of performances could be realized. Zou and his coworkers synthesized poly(naphthoxy/trifluoroetheoxy)phosphazenes (PNTFPs) with different ratios of the two side groups and controlled the carbonization temperature to obtain different carbon materials. After comparison, the material containing 4.16% atomic nitrogen content had the highest specific area of 2593 m^2^/g in 600 °C carbonization temperature, making it well suited for supercapacitor applications [142].

## 4. Conclusions

In summary, HCCP is a unique building block for synthesis of –P=N containing covalently crosslinked polyphosphazenes framework materials. The assembled polyphosphazenes particles can be well-tuned in size and morphology by varying the types of solvent and co-monomer as well as reaction conditions.

In polyphosphonitrile chemistry, chemical structures and key elements can be achieved by adjusting the two side groups connected to each phosphorus atom. This confers excellent thermal properties, flame retardant, high temperature resistance, radiation resistance and biocompatibility on such structures. Polyphosphazenes can be functionalized via bottom-up or post-synthesis approaches. The former uses functional building blocks to construct polyphosphazenes frameworks such as superhydrophobic polyphosphazenes, fluorescent polyphosphazenes and membrane materials. The latter attaches functional moieties by reacting with the surface groups of polyphosphazenes, or by doping metal or metal salts. Crosslinked polymer microspheres with surface functional groups such as carbonyl, hydroxyl and amino groups have found important applications by covalently binding antibodies, drug delivery and catalyst carriers.

Furthermore, through enrichment of heteroatoms such as P, N and O, this inorganic-organic hybrid polymer is promising as a precursor to heteroatom-doped carbon materials. Simultaneously, metal ions (such as Fe^2+^, Ag^+^, Co^2+^, Ni^2+^, Cu^2+^, etc.) are readily doped to polyphosphazenes nanoparticles through the coordination interactions with the nitrogen atoms in the P=N structures to generate a heterogeneous catalyst with enhanced electrocatalytic activity or a supercapacitor electrode material with a higher electrical ratio.

Presently, polyphosphonitrile has made great progress in functional materials, but the resultant polyphosphazenes are amorphous due to the lack of regularity and periodicity over the long-range dimensions. The formation mechanism is still not fully understood. Industrialization and large-scale production of polyphosphonitrile are still big challenges, which are important targets of polyphosphonitrile research.

## Figures and Tables

**Figure 1 polymers-15-00015-f001:**
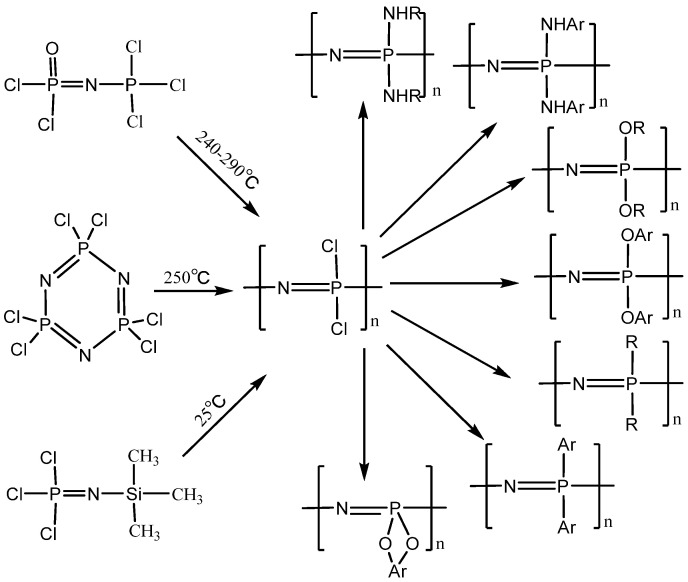
Synthetic schemes of the linear polyphosphazene derivatives. Taken with permission from [24,28].

**Figure 2 polymers-15-00015-f002:**
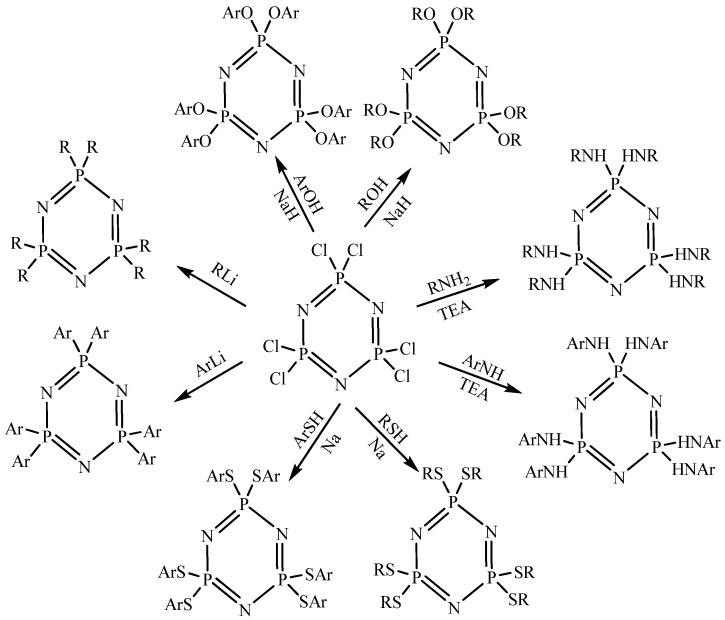
Synthetic schemes of the phosphazene derivatives. Taken with permission from [19,27].

**Figure 3 polymers-15-00015-f003:**
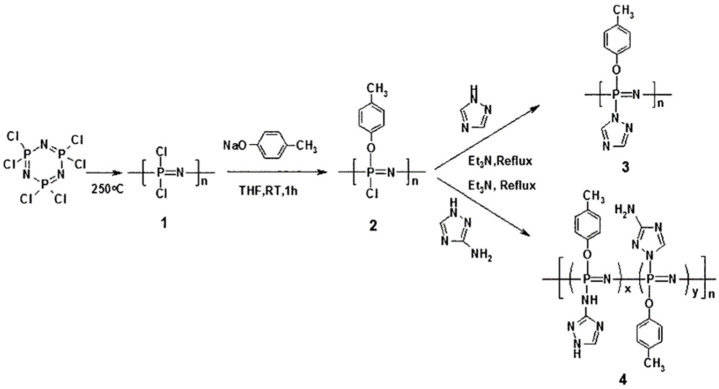
The synthetic pathway of the membrane [101].

**Figure 4 polymers-15-00015-f004:**
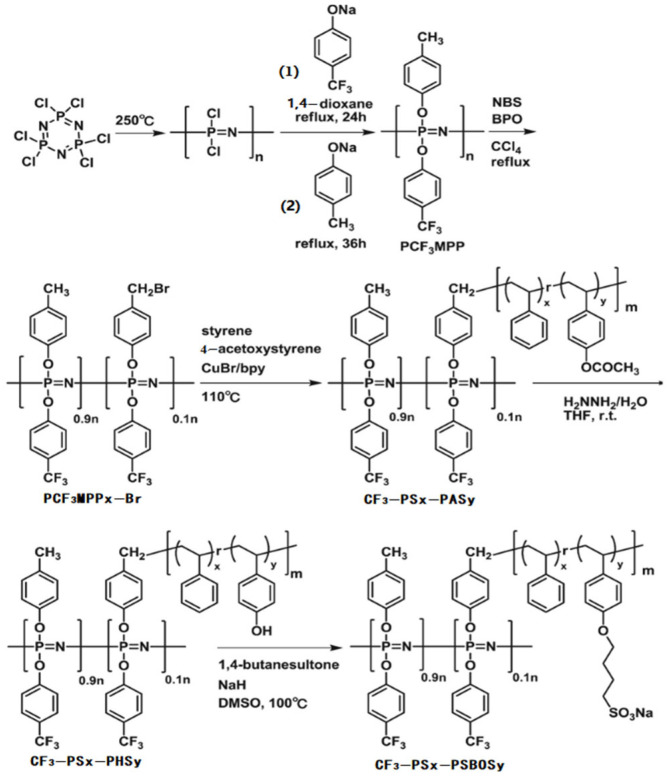
The schematic diagram of membrane preparation progress. Taken with permission from [102].

**Figure 5 polymers-15-00015-f005:**
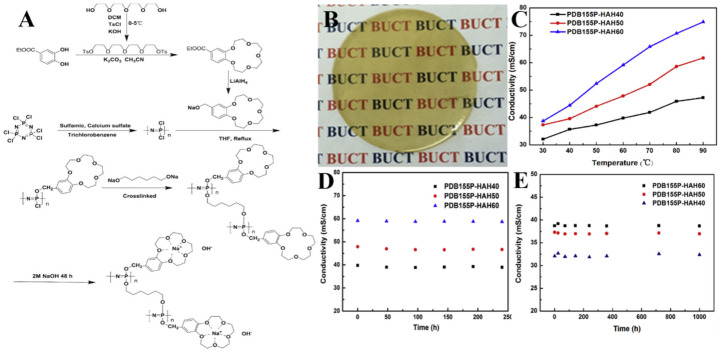
(**A**) The preparation progress of membrane. (**B**) The picture of membranes. (**C**) AEMs ionic conductivities at 30–90 °C under RH (relative humidity) of 100%. (**D**) AEMs alkaline stability in 4 mol/L NaOH at 60 °C for 10 days. (**E**) AEMs alkaline stability in 2 mol/L NaOH at room temperature for 1000 h. Taken with permission from [50].

**Figure 6 polymers-15-00015-f006:**
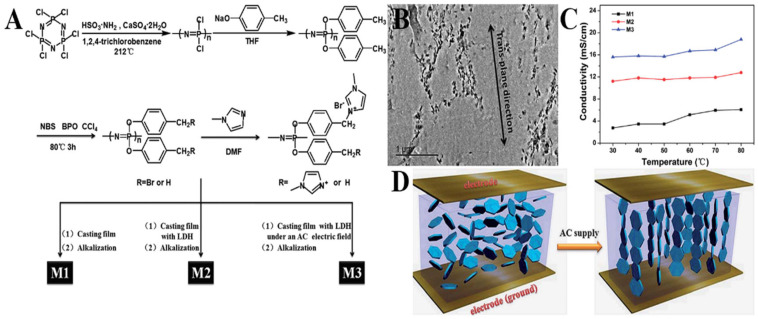
(**A**) The scheme of membrane fabrication. (**B**) TEM image of membrane with LDH under an AC electric field. (**C**) AEMs ionic conductivity as a temperature function. (**D**) The schematic diagram of electric field-introduced orientation of LDH nano-platelets in a heterogeneous solution. Taken with permission from [126].

**Figure 7 polymers-15-00015-f007:**
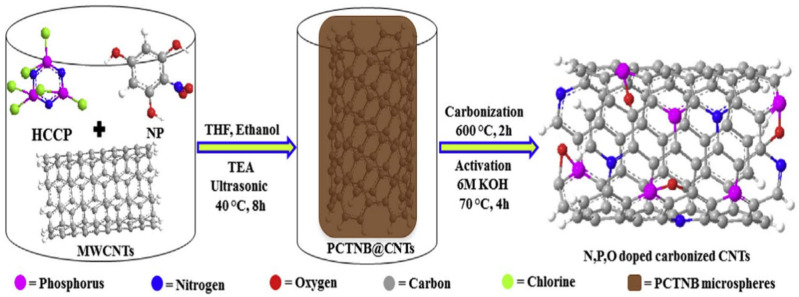
The schematic diagram of the polyphosphazenes hinged N, P, O doped carbonized PCTNB@CNTs ORR electrocatalysts. Taken with permission from [127].

**Figure 8 polymers-15-00015-f008:**
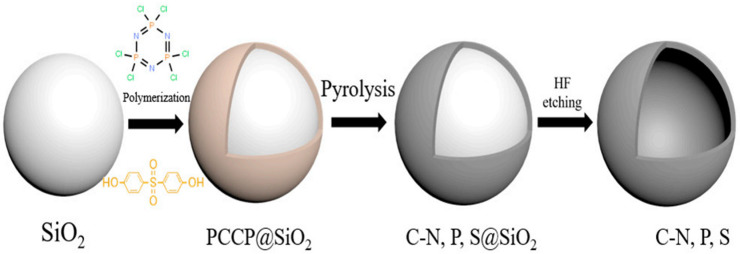
Schematic representation of hollow C–N, P, S sphere fabrication [128].

**Figure 9 polymers-15-00015-f009:**
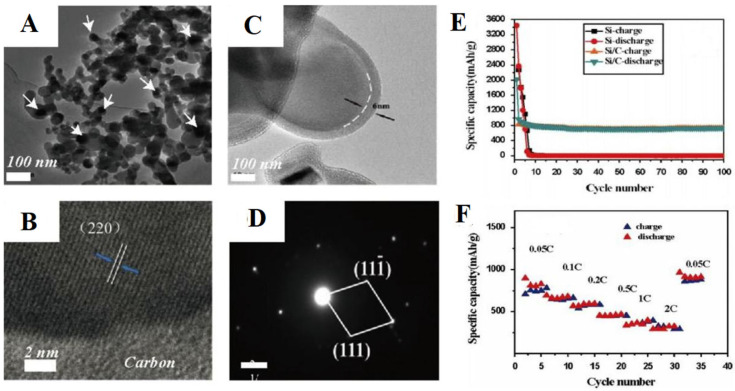
(**A**–**D**) TEM results of Si/C composite; cycling performance (**E**) and charge/discharge potentials at diverse rates (**F**) of Si/C composite. Taken with permission from [130].

**Figure 10 polymers-15-00015-f010:**
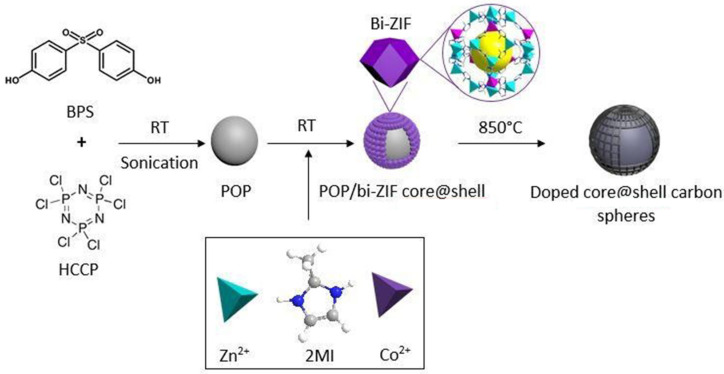
Schematic display of the synthesis along with carbonization of POP/bi–ZIFcore@shell structures. Taken with permission from [133].

**Figure 11 polymers-15-00015-f011:**
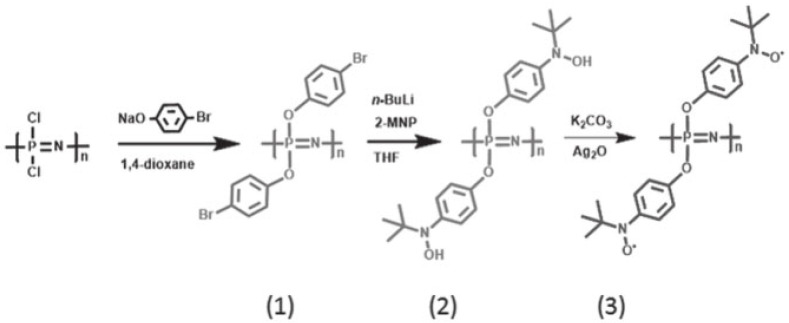
Synthetic pathway of polyphosphazenes (1–3). Taken with permission from [135].

**Figure 12 polymers-15-00015-f012:**
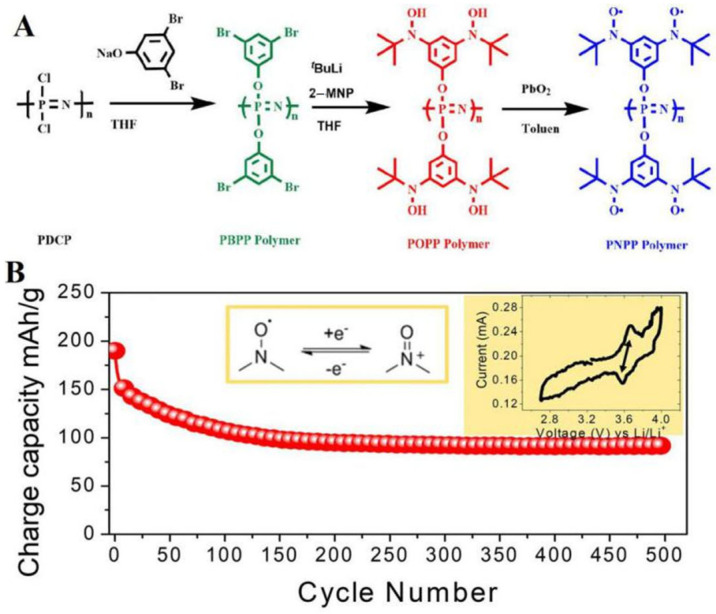
(**A**) synthesis route of the cathode-active material (PNPP); (**B**) capacity retention plot of PNPP cycled at C/2. Taken with permission from [136].

**Figure 13 polymers-15-00015-f013:**
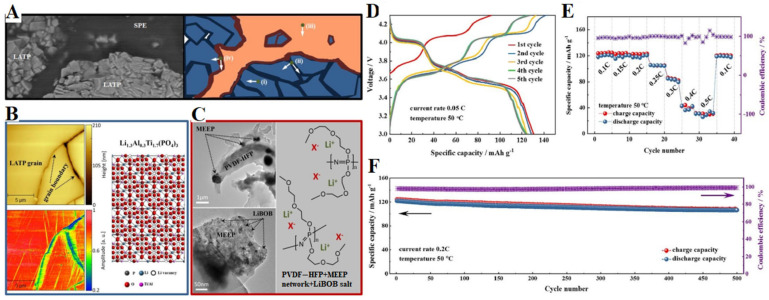
(**A**) SEM image. (**B**) amplitude signal of AFM topography along with electrochemical strain microscopy. (**C**) TEM images; (**D**) charge–discharge curve of the initial five cycles at a current rate of 0.05 °C, (**E**) rate capability at current rates from 0.1 °C to 0.5 °C, respectively, and (**F**) long–term cycling measurement at a current rate of 0.2 °C [81].

**Table 1 polymers-15-00015-t001:** The performance of polyphophazenes membranes for fuel cells.

PPNs Ion-Exchange Membranes	Ionic-Conductivity	Reference
SPOP–SPEEK	0.132 S/cm (70 °C)	[18]
M4–p-BMPP–PSx–PSBSy–4-BP	0.51 S/cm (25 °C)	[49]
PDB155P–HAH60	0.786 S/cm (90 °C)	[50]
SPEEK:SCNTB1:2	0.132 S/cm (25 °C)	[62]
PVDF-g–PSSA-0.65	0.114 S/cm (25 °C)	[92]
mPBI–TGIG–SPOP	0.143 S/cm (180 °C)	[100]
PMPCP	0.041 S/cm (130 °C)	[101]
CF_3_–PSx–PSBOSy–SCNT	0.55 S/cm (140 °C)	[102]
PPMPP-3-40%	0.223 S/cm (80 °C)	[103]
PEEK–QPOHs	0.089 S/cm (60 °C)	[104]
AEM-2	0.047 S/cm (60 °C)	[105]

## Data Availability

Further data can be made available by the corresponding author upon personal request.

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
