# Peer review of "Research Progress in Energy Based on Polyphosphazene Materials in the Past Ten Years"

_polymers, 2022, doi:10.3390/polym15010015_

Round 1
Reviewer 1 Report
Overall comments: This review introduces the research progress in the past decade on polyphosphazene in energy-related applications, primarily focusing on fuel cells and battery applications. The review gives a detailed introduction to multiple studies of polyphosphazene as different components and it can be accepted after minor revisions.
Comments:
1. The authors use the term “energy storage devices” as the general description for fuel cells and batteries. However, fuel cells are not energy storage systems but rather energy conversion systems. Please change the “energy storage devices” to “energy storage and conversion devices” accordingly when fuel cells are included.
2. The authors need to enrich the literature on the synthesis of polyphosphazenes (section 2). The current references in section 2 are between the years 1965 and 2006. Despite the focus of the manuscript being on the application of polyphosphazene in energy devices, more recent research papers and reviews need to be cited to guide the readers on the recent progress of polyphosphazene synthesis.
Some examples of the literature are:
(1) Rothemund, S. and Teasdale, I. Chem. Soc. Rev., 2016, 45, 5200-5215
(2) Gleria, M. and De Jaeger, R. Polyphosphazenes: A Review. In: Majoral, JP. (eds) New Aspects in Phosphorus Chemistry V. Topics in Current Chemistry, vol 250. Springer, Berlin, Heidelberg.
3. In addition to synthesizing polyphosphazenes via precursor (NPCl2)n, section 2 should also include the direct synthesis of poly(organo)phosphazenes.
4. The authors need to enrich the literature for section 3. For section 3.1, only four references were included for PEM applications, three references for AEM applications, and two references for catalysts for fuel cells. For section 3.2, 6 references are for electrode materials while only one reference is for polymer electrolytes in lithium batteries. More reference is needed to cover the progress of the topic in the past ten years.
5. The authors need to provide the readers with more in-depth insights into the design of polyphosphazenes for various energy-related applications. Is there any difference in the design of polyphosphazenes for membranes and catalysts? Is there any difference in the design of polyphosphazenes for different applications (fuel cells, batteries, solar cells, supercapacitors)?
Grammar comments and typos:
1. Abbreviations need to be expanded when introduced for the first time. Examples: HCCP, (NPCl2)n, etc.
2. English needs to be rechecked and improved across the manuscript.
Reviewer 2 Report
The review article looks good. However, the following are the comments.
1) Missing of line numbers and page numbers of the manuscript.
2) Provide the Scheme for the linear polyphosphazene formation by the polycondensation of N-(Dichlorophosphonyl) trichlorophosphazene and the one-step reaction using phosphorus pentachloride and ammonium chloride as monomers under the subtitle"The synthesis of polyphosphazenes".
3) There are several errors on subscript and superscript as well as symbol of centrigrade. For example, On page 3, "good selectivity of the membrane (6.08*105 S/cm3) at 60 oC, positing a promising utilization for direct methanol fuel cells. Santoshkumar and his cowork". Here cm3, 3 should be in superscript and oC should be corrected.
4) Copyright permission and citation are needed for the figures which were taken from the literature.
5) The manuscript looks like the literature review. The authors need to work hard to provide the comparative study table of the performance of polyphosphazene materials.
6) English needs to improve throughout the manuscript.
7) Rewrite the conclusions by inserting the exact findings from the discussion of the manuscript.
